# 3D Bone Morphology Alters Gene Expression, Motility, and Drug Responses in Bone Metastatic Tumor Cells

**DOI:** 10.3390/ijms21186913

**Published:** 2020-09-21

**Authors:** Ushashi C. Dadwal, Alyssa R. Merkel, Jonathan M. Page, Kristin A. Kwakwa, Michael Kessler, Julie A. Rhoades

**Affiliations:** 1Department of Veterans Affairs, Tennessee Valley Healthcare System, Nashville, TN 37212, USA; udadwal@iupui.edu (U.C.D.); alyssa.r.merkel@vumc.org (A.R.M.); kristin.a.kwakwa@vanderbilt.edu (K.A.K.); 2Department of Chemical and Biomolecular Engineering, Vanderbilt University, Nashville, TN 37235, USA; jonathanpageutk@gmail.com; 3Center for Bone Biology, Vanderbilt University Medical Center, Nashville, TN 37232, USA; 4Division of Clinical Pharmacology, Department of Medicine, Vanderbilt University Medical Center, Nashville, TN 37232, USA; 5School for Science and Math, Vanderbilt University, Nashville, TN 37235, USA; michael.r.kessler29@gmail.com; 6Department of Biomedical Engineering, Vanderbilt University, Nashville, TN 37235, USA

**Keywords:** tumor-induced bone disease, mechanotransduction, bone metastasis, 3D models, tumor microenvironment, scaffolds

## Abstract

Patients with advanced skeletal metastases arising from primary cancers including breast, lung, and prostate suffer from extreme pain, bone loss, and frequent fractures. While the importance of interactions between bone and tumors is well-established, our understanding of complex cell–cell and cell–microenvironment interactions remains limited in part due to a lack of appropriate 3D bone models. To improve our understanding of the influence of bone morphometric properties on the regulation of tumor-induced bone disease (TIBD), we utilized bone-like 3D scaffolds in vitro and in vivo. Scaffolds were seeded with tumor cells, and changes in cell motility, proliferation, and gene expression were measured. Genes associated with TIBD significantly increased with increasing scaffold rigidity. Drug response differed when tumors were cultured in 3D compared to 2D. Inhibitors for Integrin β3 and TGF-β Receptor II significantly reduced bone-metastatic gene expression in 2D but not 3D, while treatment with the Gli antagonist GANT58 significantly reduced gene expression in both 2D and 3D. When tumor-seeded 3D scaffolds were implanted into mice, infiltration of myeloid progenitors changed in response to pore size and rigidity. This study demonstrates a versatile 3D model of bone used to study the influence of mechanical and morphometric properties of bone on TIBD.

## 1. Introduction

Tumor-induced bone disease (TIBD) is a common occurrence among primary breast, lung, and prostate cancer patients, with up to 80% of patients who die from the disease displaying bone metastases upon post-mortem analysis [1]. Bone metastases lead to chronic bone pain, hypercalcemia, osteolytic lesions, pathological fractures, and bone loss, which significantly decrease patient quality of life [2]. Prior studies have demonstrated that in the primary tumor, mechanical forces exerted on tumor cells by the microenvironment promote integral changes in cell morphology, gene expression, cellular metabolism, protein synthesis, and response to cytotoxic therapeutics [3,4]. The unique architecture of trabecular bone contributes to mechanical forces on cells resulting from the mineralized extracellular matrix [5,6], curved trabecular surfaces [7,8,9,10,11], interstitial fluid flow [12,13,14], and dynamic compressive loading [15]. These bone-specific mechanical forces can affect the colonization and establishment of tumor cells in bone. While many research groups have studied the molecular events leading to tumor establishment and bone destruction, the specific cellular and mechanical cues that govern these responses have not been explored in the context of the 3D bone-tumor microenvironment [16,17].

Biomaterials mimicking the bone-tumor microenvironment have been highlighted as critical for understanding the dynamics of tumor progression, recapitulating in vivo conditions at distinct steps of metastasis, and determining how tumor cells integrate mechanical and chemical signals over multiple length scales [18]. Biomimetic 3D tissue-engineered scaffolds have been proposed as alternatives to 2D culture and mouse models for investigating molecular mechanisms of disease progression and for screening drugs [17,18,19,20]. Hydroxyapatite [21], collagen [22,23], poly(lactic-co-glycolic acid) (PLGA) [24], and poly(ε-caprolactone) (PCL) [25] scaffolds have been investigated as 3D models for assessing tumor cell gene expression and drug response in TIBD. However, the ability of electrospun polymeric scaffolds and hydrogels to recapitulate the mechanical and morphometric properties of the bone microenvironment may be limited by their low substrate modulus and nanoscale pore size. Consequently, the effects of mechanical and morphometric properties on bone-metastatic gene expression and drug response have not been systematically investigated for scaffolds with substrate moduli and pore sizes representative of trabecular bone.

In this study, we designed 3D-printed scaffolds to investigate the effects of mechanical forces in the bone microenvironment on progression of TIBD. We previously reported that the substrate modulus of 2D films regulates tumor cell expression of the transcription factor Gli2 and parathyroid hormone-related protein (*PTHrP*), both of which have been associated with bone destruction [5,6]. However, in the 3D bone microenvironment, cells are exposed to mechanical forces resulting from surface curvature [26] in addition to the substrate modulus, which stimulates mechanically transduced signaling in tumor [6] and bone [27,28] cells. To more accurately recapitulate the mechanical and morphometric properties of the bone microenvironment, we fabricated 3D scaffolds with tunable substrate moduli and pore size using a templated-Fused Deposition Modeling (3D t-FDM) process [28,29]. The bulk substrate modulus ranged from 5 to 266 MPa, which is representative of values reported for collagen fibers (32 MPa) and trabecular bone (93–366 MPa). Pore sizes in the physiologically relevant range that supports bone formation and vascularization (300–600 µm) were also evaluated [5,27,28,29,30]. Scaffolds were seeded with bone-metastatic tumor cells to investigate the effects of substrate modulus and pore size on bone-metastatic gene expression and drug response in vitro using multiple cancer cell lines, and in vivo using a breast cancer xenograft model.

## 2. Results

### 2.1. 3D Scaffolds Recapitulate the Mechanical and Morphometric Properties of Trabecular Bone

We fabricated 3D scaffolds designed to recapitulate both bone-specific mechanical forces arising from the rigid extracellular matrix and macropores between curved trabeculae. Two different substrate moduli were used: (i) compliant (C, 5 MPa) scaffolds that are less rigid than a collagen fiber (32 MPa), and (ii) rigid (R, 266 MPa) scaffolds that are representative of trabecular bone (93–365 MPa) [5,28,30]. Pore sizes were selected to be within the range that supports bone formation and vascularization (>300 μm). The smaller pore size (420 μm) is within the optimal range for osteogenesis [31,32,33,34], while the larger pore size (560 μm) is representative of tumor-bearing trabecular bone (Trabecular Separation, Tb.Sp.) ranging from 430 to 930 μm [28,30,32]. Four scaffolds were fabricated: (1) compliant scaffolds with 560 μm pores (560C), (2) compliant scaffolds with 420 μm pores (420C), (3) rigid scaffolds with 560 μm pores (560R), and (4) rigid scaffolds with 420 μm pores (420R). Representative images of the scaffolds are in Figure 1B and scaffold properties are listed in Table 1.

### 2.2. 3D Scaffolds Support Proliferation of Tumor Cells

Scaffolds were seeded with bone-tropic MDA-MB-231 tumor cells, and cells were evaluated for changes in gene expression, cell growth, and motility in response to varying pore sizes and rigidity. Viability of MDA-MB-231 and patient-derived chondrosarcoma cells assessed by Trypan blue staining demonstrated < 5% cell death after 48 h, indicating that the 3D scaffolds are not cytotoxic (Appendix A). Scanning electron microscopy (SEM) images show that the tumor cells spread on and adhere to the surface of all four scaffold groups (Figure 1C). Previous studies have shown expression of Collagen-1 and Fibronection in MSCs cultured on 3D scaffolds [28]. Here, we demonstrate changes in expression of proteins associated with tumor attachment and adhesion to the environment including Focal Adhesion Kinase (FAK) and Integrin Beta 3 (Iβ3). FAK expression increases on MDA-MB-231 cells grown on rigid scaffolds compared to compliant scaffolds as shown by increased number of focal adhesions per cell (Appendix A). We further investigated whether pore size and substrate modulus affected cell proliferation from day 1 to 3. Cell proliferation as measured by increases in cell number (GFP fluorescence expression) or metabolic activity (MTS assay) was not significantly affected by substrate modulus or pore size (Figure 1D,E).

### 2.3. Substrate Modulus and Pore Size Regulate the Motility of Tumor Cells

Since previous studies have established that the microenvironment can alter cell motility and invasive potential [16,17,35,36], we investigated the effects of substrate modulus and pore size on motility and migration of bone-metastatic MDA-MB-231-GFP tumor cells cultured on 2D and 3D substrates. 2D films with 5 MPa (2D Compliant) or 266 MPa (2D Rigid) bulk substrate moduli were synthesized from the same components as the 3D scaffolds to minimize the effects of chemical composition on cell fate (Figure 2A). Using live cell tracking, we found that tumor cells were significantly more motile on 3D scaffolds compared to 2D films (Appendix A). We observed 6- and 7-fold higher cell displacements for the 560R and 420R scaffolds compared to the 2D compliant films (Figure 2B). When tumor cell-seeded scaffolds were placed in serum-free media for a period of 6 h and exposed to 10% FBS media (chemoattractant, CM), cell speed increased significantly (5-fold 560R and 4-fold 420R) with increasing substrate rigidity but was independent of pore size (Figure 2C).

To determine if this increased motility can result in an increase in tumor migration, we placed tumor seeded scaffolds on top of a transwell and measured the number of cells that migrated through. Transwell migration assays demonstrated significantly higher migration potential of cells on 420R (≈3-fold) and 560R (≈2.5-fold) scaffolds compared to compliant scaffolds in complete media (CM), while there were no significant changes in migration potential without a chemoattractant gradient (SFM) (Figure 2D). Taken together, these observations suggest that the rigidity of the 3D microenvironment, but not pore size, can increase cell motility.

### 2.4. 3D Scaffolds Influence the Expression of Bone-Metastatic Genes In Vitro

To test the effects of substrate modulus and pore size, both parameters that influence mechanical signaling, on gene expression in bone-metastatic tumor cells, MDA-MB-231, RWGT2, and PC3 cells were seeded onto 2D films or 3D scaffolds and cultured for 48 h. Bone-metastatic gene expression was analyzed by qRT-PCR. *ITGB3*, *Gli2*, and *PTHrP* expression were significantly affected by both increasing rigidity and alterations in pore size. *ITGB3* expression was 10-fold higher in MDA-MB-231 cells, 5–7-fold higher in RWGT2 cells, and 5–10-fold higher in PC3 cells on rigid compared to compliant scaffolds (Figure 3A). Furthermore, there was a 2-fold significant increase in *ITGB3* gene expression in PC3 cells grown in a 560 μM scaffold compared to 460 μM scaffolds or 2D films. *Gli2* expression significantly increased almost 2-fold with increasing rigidity and decreasing pore size in MDA-MB-231 cells, while *Gli2* expression was highest (2-fold) in 560R scaffolds for RWGT2 and PC3 cells (Figure 3B). *PTHrP* expression was 10-fold higher in MDA-MB-231 and RWGT2 cells, and 3-fold higher in PC3 cells on rigid compared to compliant scaffolds (Figure 3C). *PTHrP* increased with decreasing pore size in MDA-MB-231 cells but pore size differences were not observed for RWGT2 and PC3 cells. These data suggest that substrate modulus and pore size regulate expression of genes associated with bone metastasis in breast cancer (MDA-MB-231), lung cancer (RWGT2), and prostate cancer (PC3).

### 2.5. 3D Scaffolds Influence the Response of Tumor Cells to Therapeutics

To further explore the effect of the 3D bone microenvironment on tumor cell behavior, we tested the drug response of the tumor cell lines to three inhibitors in short-term mono-culture on 2D vs. 3D rigid scaffolds. MDA-MB-231, RWGT2, and PC3 cells were cultured on rigid 2D films or 420R 3D scaffolds and expression of *ITGB3, GLI2*, and *PTHrP* was measured by qRT-PCR after 48 h of drug treatment. The TGF-β Receptor I kinase inhibitor (SD-208) and the Integrin inhibitor (Cilengitide, Cil) significantly reduced expression of genes associated with TIBD in 2D films by 2–3-fold; however, these drugs were less or not effective in 3D scaffolds (Figure 4A,B). In contrast, treatment with the Gli antagonist GANT58 both consistently and significantly reduced bone-metastatic gene expression >3-fold in all three cell lines and in both culture conditions. Similar experiments were performed on compliant films and scaffolds; however, low expression caused a decrease in yield. In support, molecular inhibition of these pathways in MDA-MB-231 cells using a mutant TGF-β receptor type II construct (rII Δcyst) [37], shRNA for integrin β3 (shβ3), or a Gli2 repressor construct (Gli2 EnR) reduced gene expression in both 2D and 3D (Appendix A). Taken together, these data reveal that the characteristics of the bone microenvironment like rigidity and 3-dimensional structure mediate the efficacy of therapeutics, highlighting the importance of 3D culture systems when studying bone-metastatic tumor cell response.

### 2.6. Substrate Modulus and Pore Size Mediate Expression of Bone-Metastatic Genes in a Breast Cancer Xenograft Model

In order to determine how the 3D microenvironment regulates tumor growth in vivo, we implanted MDA-MB-231-seeded 3D scaffolds in the abdomen of athymic nude mice. After 3.5 weeks, scaffolds were harvested for histomorphometry analysis or gene expression analysis (Figure 5A). Expression of genes associated with TIBD increased with greater substrate modulus and decreasing pore size (Figure 5B–D). *ITGB3*, *GLI2*, and *PTHrP* mRNA significantly increased 20-, 18-, and 35-fold on 420R compared to 560C scaffolds. Immunohistochemistry for the proliferation marker Ki67 and the apoptosis marker Caspase 3 (Casp3) show increased cell proliferation on 420R scaffolds and little apoptosis activity on all scaffolds (Appendix A). We also determined via a Tumor Metastatic panel several genes associated with bone remodeling were highly expressed in the 3D rigid scaffolds when compared to the complaint scaffolds (Appendix A). These data suggest that these 3D scaffolds could be a powerful tool for studying tumor growth and response to therapeutics in vivo.

### 2.7. Substrate Modulus Influences Immune Cell Infiltration

Previous studies have emphasized the contributions of the host microenvironment and immune response to tumor behavior during tumor establishment, proliferation, and growth [38]. We investigated whether the host immune cell populations that infiltrated the scaffolds created a pro-tumorigenic niche by examining the myeloid (CD11b+), macrophages (F4/80+), and granulocytic (Ly6G+) immune cell populations that are known to support TIBD [39,40,41]. We confirmed a significant 6-fold increase in the population of CD11b+ cells and a 10-fold increase in F4/80+ cells in 420R compared to 560C 3D scaffolds (Figure 6A–D). Similar to the bone-metastatic gene expression data, the population of CD11b+ and F4/80+ cells increased with increasing substrate modulus and decreasing pore size. We also detected the presence of Ly-6G+ positive cells in all scaffolds (Appendix A). This data shows that the 3D scaffolds recapitulate the tumor–immune cell interactions in the bone microenvironment during TIBD, demonstrating that the 3D scaffolds are a robust and reliable model to study components of bone metastatic disease.

## 3. Discussion

In the present study, we investigated the effects of substrate modulus and pore size (surface curvature) on tumor cell gene expression and drug response using 3D scaffolds with mechanical and morphometric properties comparable to trabecular bone (Figure 7). Expression of genes associated with TIBD increased with increasing substrate modulus and decreasing pore size (increasing surface curvature). Furthermore, while the Gli antagonist GANT58 was effective in both 2D and 3D culture, Iβ3 and TGF-β inhibitors were not as effective on 3D bone-like scaffolds. These findings underscore the importance of screening tumor-targeting therapies in a microenvironment that is representative of host tissue.

Tissue-engineered 3D scaffolds are a promising technology for independently controlling the mechanical properties of the tissue microenvironment [18]. Previous studies using 3D scaffolds to model bone-metastatic breast cancer [5,42,43], Ewing’s sarcoma [25], and bone-metastatic prostate cancer [22,44] have reported significant differences between 2D and 3D culture. Ewing’s sarcoma cells cultured on 3D electrospun PCL scaffolds showed higher expression and activation of insulin-like growth factor-1 receptor (IGF-1R), as well as higher expression of proteins associated with resistance to IGF-1R therapy, compared to 2D mono-culture [25]. In another study, prostate cancer cells cultured in 3D hydroxyapatite (HA)-collagen scaffolds exhibited reduced expression of MMP1 and MMP9 compared to 2D culture [22]. Similarly, we have previously reported that expression of MMP9 by MDA-MB-231 cells cultured on 2D films decreased with substrate rigidity [6]. Coculture of prostate cancer cells with human osteoblasts on poly(ε-caprolactone) (PCL)-tricalcium phosphate (TCP) scaffolds exhibited upregulation of matrix metalloproteinases (MMPs) and prostate specific antigen (PSA) compared to prostate cancer cells alone [44]. While these studies highlight the substantial contribution of the 3D microenvironment to tumor cell fate, the relative contributions of mechanical forces resulting from substrate rigidity and surface curvature on tumor cell gene expression in bone have not been systematically investigated.

The complexity and spatial heterogeneity of the bone microenvironment [28,35,45,46] has made it difficult to predict which tumors will colonize the bone and induce bone destruction. Recently, we developed fabricated tissue-engineered bone constructs (TEBCs) from human trabecular bone templates that are capable of mimicking human bone morphometric properties—trabecular interconnectivity, surface roughness, and mechanical properties to study human mesenchymal stem cell (hMSCs) regulation of proliferation, differentiation, and mineralization [11]. Our previous study demonstrated hMSCs cultured on TEBCs exhibited significantly different metabolic activities, osteogenic differentiation, and mineralization depending on the anatomic site. Although many approaches to 3D models of bone and tumor exist, by using high throughput and tunable 3D scaffolds presented in this study we demonstrate a faster, more readily available, and more affordable 3D scaffold which is user-friendly to molecular biology laboratories.

Our studies demonstrated that these 3D scaffolds in vitro showed an important distinction between 2D and 3D culture. This difference was further exemplified while testing inhibitors that target components of TIBD. We also demonstrate in vivo that the 3D scaffolds reflected critical components of the tumor-bone microenvironment including immune cell infiltration, tumor proliferation, and pro-TIBD gene expression. Although these studies were performed with cells that primarily form osteolytic lesions, the 3D scaffolds described herein also have potential to be used to study tumors such as prostate cancer that form osteoblastic lesions. Based on these studies, while this model does not fully recapitulate bone, it improves upon tissue culture plastic or simple Matrigel studies to be a more bone mimicking in vitro microenvironment. For many in vitro molecular biology and in vivo cellular studies, this simpler approach is sufficient. However, we recognize that other approaches may still require more bone-like scaffolds like the TEBCs.

Trabecular bone comprises a complex mixture of rod and plate-like trabeculae with an average spacing of 600–800 µm apart and substrate modulus 93–365 MPa, which is several orders of magnitude higher than that of soft tissue [5,6,39,47]. Tumor cells in the bone microenvironment are subject to increased mechanical forces compared to soft tissue due to the mineralized extracellular matrix, curved trabeculae, and fluid shear stress. Consistent with our previous study using 2D films [5], expression of *ITGB3*, *GLI2*, and *PTHRP* by tumor cells cultured on 3D scaffolds increased with increasing substrate modulus due to interactions between integrins and growth factors. Similarly, we have reported that osteoblast differentiation of mesenchymal stem cells increased with increasing substrate modulus and decreasing pore size due to physical association of Iβ1 and BMP Receptor I (BMPRI) [30]. These findings are consistent with previous studies reporting that the magnitude of cellular contractile forces increases with increasing surface curvature [48,49]. Since trabecular separation decreases with proximity to the cortex, our findings suggest that tumor cells may become more invasive as they migrate toward the cortical wall.

Previous studies have reported that antitumor drugs are not as effective in the 3D bone microenvironment compared to 2D culture [17,22,25,50]. Therefore, we tested several inhibitors of TIBD in 2D and 3D culture to identify differences in drug response: Cilengitide (Iβ3 inhibitor), SD-208 (TGF-β Receptor I kinase inhibitor), and GANT58 (Gli antagonist). Cilengitide and SD-208 significantly reduced expression of *ITGB3*, *GLI2*, and *PTHRP* in 2D culture, but were less effective at reducing expression of these genes in 3D culture. While TGF-β inhibitors block metastasis to bone and subsequent bone destruction [51,52], they are less effective when given after tumors have established in bone [38]. However, it is well-established that TGF-β inhibitors increase bone volume independent of tumor [53,54] and thus would be beneficial even when given after tumors have established in bone. Iβ3 inhibitors have also been reported to block tumor metastasis to bone [52,55], as well as directly target osteoclasts and prevent bone resorption [56]. Our findings suggest that once tumors are established in bone, they are subjected to mechanical forces significantly greater than in the primary site and this increased signaling activation may reduce their ability to respond to Iβ3 and TGF-β inhibitors. In contrast, GANT58 reduced expression of *ITGB3*, *GLI2*, and *PTHRP* in tumor cells cultured on both 2D films and bone-like 3D scaffolds, which highlights the potential of Gli2 inhibitors for blocking TIBD. It was recently shown that targeting Gli2 inhibition to the tumor-bone microenvironment effectively reduced TIBD in a model of breast cancer metastasis to bone [57,58]. Thus, this study emphasizes the importance of studying tumor cell signaling and response to therapeutics in 3D cultures that accurately capture the physical properties of the bone microenvironment.

## 4. Materials and Methods

### 4.1. Cell Lines

A bone metastatic clone of the human breast cancer cell line MDA-MB-231 was derived by our lab as previously described [59]. The human squamous lung cancer cell line, RWGT2, was derived as previously described [60]. The human prostate cancer cell line, PC3, was purchased from ATCC (Manassas, VA, USA). MDA-MB-231-GFP cells were generated by transfecting a bone-metastatic variant of MDA-MB-231 cells with a Green Fluorescence Protein (GFP) overexpression plasmid and passaged through mice to further select for a bone metastatic clone [5,61,62]. MDA-shβ3, MDA-Gli2 EnR, and MDA-rII ΔCyst cells were generated for molecular inhibition experiments. Detailed methods can be found in supplementary methods section. Cells were tested every 6 months using the MycoAlert Mycoplasma Detection Kit (Lonza, Alpharetta, GA, USA) and every 2 years by the Vanderbilt Translational Pathology Shared Resource for multiple infectious agents.

### 4.2. Synthesis of 2D Poly(Ester Urethane) (PEUR) Films

2D poly(ester urethane) (PEUR) films were synthesized by reacting a mixture of poly(ε-caprolactone-*co*-glycolide) triol (*M*_n_ = 300 or 3000 g mol^−1^), hexamethylene diisocyanate trimer (HDIt), and stannous octoate catalyst as described previously [5]. To facilitate cell adhesion, fibronectin (Fn) was adsorbed to the surface of the substrates by incubating them in a 4 µg mL^−1^ solution of Fn in PBS at 4 °C overnight.

### 4.3. Fabrication and Characterization of 3D Scaffolds by templated-Fused Deposition Modeling (t-FDM)

3D scaffolds were fabricated as described previously (Figure 1A) [28,30]. Briefly, a poly(lactic acid) (PLA) template was printed using a MakerBot Replicator^®^ 2 Fused Deposition Modeling (FDM) (Brooklyn, NY, USA) printer and filled with the PEUR mixture listed above. The PLA template was leached with dichloromethane (DCM) to yield 3D scaffolds with interconnected pores having a diameter of 423 ± 34 or 557 ± 44 μm for the nominal 420 or 560 μm templates, respectively. Pore size and morphology were characterized by scanning electron microscopy (SEM) (Hitachi S-4200, Finchampstead, Royal Bekshire, UK). Scaffolds were sterilized under UV light for 15 min in 70% ethanol and incubated in a solution of 4 μg mL^−1^ fibronectin overnight at 4 °C. As we have reported previously [28], the bulk substrate modulus (*K*_s_, measured by compression testing) of the 3D scaffolds was controlled by the molecular weight (*M*_w_) of the polyester triol (3000 or 300 g mol^−1^) to attain values representative of collagen (5 MPa, Compliant (C)) or trabecular bone (266 MPa, Rigid (R)) (Table 1).

### 4.4. Cell Culture

MDA-MB-231 cells were cultured in DMEM; RWGT2 cells were cultured in α-MEM; PC3 cells were cultured in RPMI growth media. All media was supplemented with 10% FBS (Peak Serum, Wellington, CO, USA) and 1 % Penicillin streptomycin solution (Mediatech, Manassa, VA, USA). 2D films were seeded with 3.6 × 10^5^ tumor cells and 3D scaffolds were seeded with 0.5 × 10^6^ cells/scaffold.

### 4.5. Cell Viability, Proliferation, and Metabolic Activity

Trypan Blue (Sigma, St. Louis, MO, USA) was used to measure cell viability 48 h after seeding (detailed methods can be found in Appendix A). Metabolism was measured by MTS assay (CellTiter 96^®^ Aqueous Non-Radioactive Cell Proliferation Assay, Promega, Madison, WI, USA) per the manufacturer’s protocol after 1, 2, and 3 days in culture (Figure 1D). Cell proliferation was measured by fluorescence intensity measurement in MDA-MB-231-GFP cells after 1, 2, and 3 days in culture using a Synergy 2 plate reader (Biotek Instruments, Winooski, VT, USA) (Figure 1E).

### 4.6. Cell Migration Assays

MDA-MB-231-GFP cells were plated on fibronectin-coated 2D films or 3D scaffolds, placed in a live cell chamber (LiveCell™, Okolab, Ambridge, PA, USA) at 5% CO_2_ and 37 °C and monitored by light microscopy (Olympus CKX41, Center Valley, PA, USA). Images of the same field were taken every hour for 48 h, and ImageJ software (NIH, Bethesda, MD, USA) [63] was used to analyze the photo series to track single cell (*n* = 3) movement. Chemotactic migration was quantified using a 24-well plate Boyden chamber transwell assay (8 μm pore size; Corning Costar, Tewksbury, MA, USA). 3D scaffolds were seeded with 4.0 × 10^4^ tumor cells, serum-starved for 4 h, and mounted into the upper chamber of the transwell. Medium in the lower chamber was supplemented with 10% (positive control) or 0% (negative control) FBS. After 30 h, cells present on the upper surface of the filter were removed using a sterile cotton swab, and cells that migrated through the chamber onto the lower surface were fixed and stained with Crystal Violet Dye (Sigma, St. Louis, MO, USA). The number of migrating cells was counted (*n* = 5 fields) and compared to the negative control. Experiments were performed in triplicate wells and repeated at least three times.

### 4.7. Scanning Electron Microscopy

3D scaffolds were seeded with tumor cells and fixed with 4% paraformaldehyde for 1 h. Subsequently, samples were further fixed in a solution of 5% glutaraldehyde, followed by 2% osmium tetraoxide. Samples were dehydrated using ethanol washes of increasing concentration and vacuum dried overnight. Once fixed and dried, 3D scaffolds were cut open to expose pores and mounted on a stub using carbon tape. Samples were sputter-coated with gold using a 108 Auto Sputter Coater (Ted Pella Inc., Redding, CA, USA) and imaged on a Zeiss Merlin scanning electron microscope (Carl Zeiss Inc., Thornwood, White Plains, NY, USA).

### 4.8. Drug Treatments

2D films or 3D scaffolds were seeded with tumor cells, cultured for 24 h, and then treated with 10 μM Cilengitide (Selleck Chemicals, Huston, TX, USA), 10 μM GANT 58 (Santa Cruz Biotechnology, Dallas, TX, USA), 100 nM SD208 (Sigma-Aldrich, St. Louis, MO, USA), or DMSO as a control in serum-free media. After 48 h of drug treatment, RNA was harvested for qRT-PCR analysis.

### 4.9. Quantitative RT-PCR

To measure changes in gene expression, quantitative mRNA reverse transcription polymerase chain reaction was carried out using a TaqMan^®^ gene expression assay. Briefly, cells were harvested with TRIzol (Invitrogen, Waltham, MA, USA) after 48 h in culture and total RNA was extracted following manufacturer’s instructions. The qScript cDNA supermix (Quanta, Beverly, MA, USA) was used to synthesize cDNA using 1 μg total RNA. Using validated TaqMan^®^ primers, the expression of *PTHLH* (Hs00174969_m1), *Gli2* (Hs01119974_m1), and *ITGB3* (Hs01001469_m1) was measured in triplicate by quantitative RT-PCR with the 7500 Real-Time PCR System (Applied Biosciences, Waltham, MA, USA) under the following cycling conditions: 95 °C for 10 min, followed by 40 cycles of 95 °C for 15 s and 60 °C for 1 min. mRNA concentration was calculated against a standard curve using the absolute quantification method. 18SrRNA was used as an internal control.

### 4.10. In Vivo Studies

All animal procedures were approved by the Vanderbilt University Institutional Animal Care and Use Committee (IACUC, protocol M/06/213, approval date: 22 August 2013) and were conducted according to NIH guidelines. For in vivo studies, 1 × 10^6^ MDA-MB-231-GFP cells were seeded into 3D scaffolds. For subcutaneous implants, female 4-week-old athymic nude mice (*Mus musculus* Hsd:Athymic Nude-*Foxn1^nu^*, Envigo, Indianapolis, IN, USA) were anesthetized by continuous isoflurane and an incision made on the ventral lower abdomen. The left inguinal mammary gland between the fourth and fifth mammary fat pad was transplanted with cell-seeded 3D scaffolds. Mice were imaged weekly for fluorescence using the MAESTRO™ (CRi, Hopkins, MA, USA) imaging system. The collected images were spectrally unmixed to remove background fluorescence. Mice were sacrificed at 3.5 weeks post-tumor cell inoculation and the tumors excised and processed for gene expression analysis or histomorphometry.

### 4.11. Histomorphometry

Excised tumors were formalin fixed for 48 h and processed for paraffin embedding. Then, 4 µm sections were cut using a Leica RM2255 (Buffalo Grove, IL, USA) and deparaffinized in xylenes and stained with Hematoxylin and Eosin for histological analyses. Immunohistochemical staining was performed to analyze infiltration of immune cells (CD11b, F4/80, Ly6G) and cellular proliferation (Ki67/Casp3). Detailed methods can be found in the supplemental methods section. All quantitative analyses were performed using Metamorph Analysis (Meta Imaging) or ImageJ software (NIH, Bethesda, MD, USA).

### 4.12. Statistical Analysis

All studies were performed in triplicate with at least *n* = 3 test substrates per replicate to ensure batch-to-batch reproducibility. ANOVA and the Tukey’s multi comparison post hoc test were used for statistical analyses with significance set at *p* < 0.05 unless otherwise stated. Cell culture data was tested for significance across time points, across materials, and interactions between time and materials with *n* = 3 independent biological experiments.

## 5. Conclusions

This study showed how substrate modulus and pore size affects tumor cell behavior using a tunable 3D culture system. We found that tumor cells grown in a more rigid environment with a smaller pore size were more motile and had amplified expression of genes associated with bone metastasis, and that the drug response in a 3D bone-like environment differed remarkably from 2D monoculture. Furthermore, this 3D system can be used in an in vivo orthotopic model to recapitulate the tumor-bone microenvironment. This 3D bone-like model will potentially enable screening of new therapeutics both in vitro and in vivo prior to preclinical testing.

## Figures and Tables

**Figure 1 ijms-21-06913-f001:**
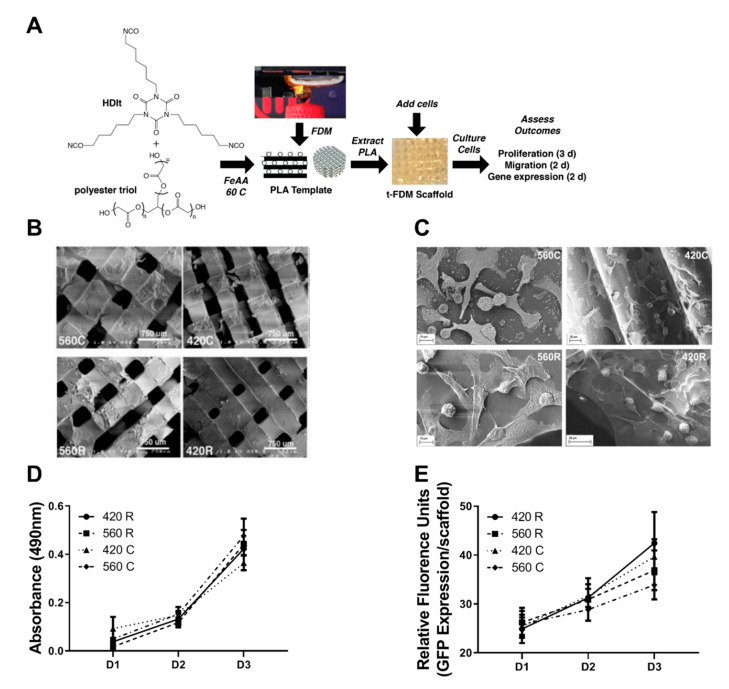
Fabrication schematic for 3D t-FDM scaffolds and cellular activity on the scaffolds. (**A**) Hexamethylene diisocyanate trimer (HDIt) was mixed with a poly(ε-caprolactone-co-glycolide) triol and stannous octoate catalyst and poured into a poly(lactic acid) (PLA) mold fabricated by a MakerBot Replicator 2 fused-deposition modeling (FDM) 3D printer. After curing overnight, the PLA mold was extracted via solvent leaching in dichloromethane for 24 h. The resulting scaffold was washed with DI water and prepared for cell culture. (**B**,**C**) Representative scanning electron microscopy (SEM) image of the 3D t-FDM scaffolds (**B**) and of cell attachment on 3D scaffolds (**C**)—560 µm compliant (560C), 420 µm compliant (420C), 560 µm rigid (560R), and 420 µm rigid (420R). (**D**) Metabolic activity of MDA-MB-231 tumor cells cultured on 3D scaffolds measured by the MTS assay corroborates no variation of metabolic expression over a time period (days- D1, D2 and D3). (**E**) Proliferation of MDA-MB-231 tumor cells on 3D scaffolds measured as fluorescence expression of GFP, demonstrates no statistical differences between 3D scaffold groups over a time period (days- D1, D2 and D3).

**Figure 2 ijms-21-06913-f002:**
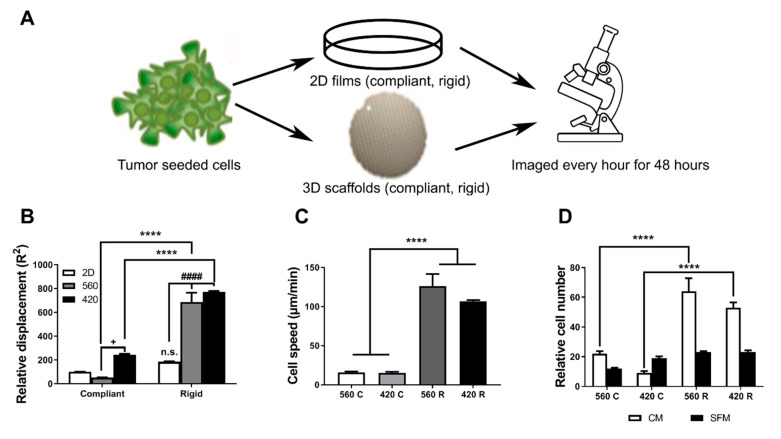
Rigidity and pore size affect migration of tumor cells on 3D scaffolds. (**A**) Schematic of experimental procedure (**B**) Live cell imaging using bright field microscopy demonstrates significantly higher cell motility on 560R (>6-fold) and 420R (>7-fold) scaffolds as compared to compliant films (2D Compliant, white column). (**C**) Motility assay using confocal imaging shows that tumor cells have significantly increased speed on the 560R (>8-fold) and 420R (>7-fold) scaffolds compared to compliant scaffolds. (**D**) 3D migration transwell assay with either 10% FBS media (CM) or serum-free medium (SFM) determined tumor cells have higher migratory potential on 560R (3-fold) and 420R (>2.5-fold) 3D scaffolds as compared to 560C and 420C scaffolds. 2way ANOVA. Compliant vs. Rigid, **** *p* < 0.0001. 2D vs. 3D, #### *p* < 0.0001. 560 vs. 420, + *p* < 0.05.

**Figure 3 ijms-21-06913-f003:**
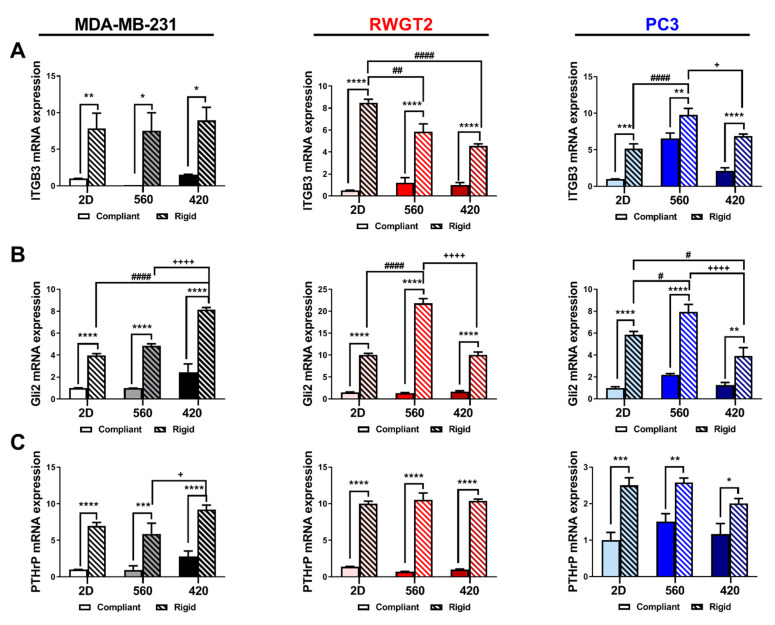
Effects of substrate modulus and pore size on gene expression of bone-metastatic tumor cells. The breast cancer cell line, MDA-MB-231 (black), the lung cancer cell line, RWGT2 (red), and the prostate cancer cell line, PC3 (blue), were seeded on 2D films or 3D scaffolds, cultured for 48 h and analyzed for changes in gene expression. Expression of (**A**) ITGB3, (**B**) Gli2, and (**C**) PTHrP were significantly increased for all cell types analyzed with respect to changes in both pore size and rigidity. Data presented as fold change over 2D compliant. Two-way ANOVA. Compliant vs. rigid, * *p* < 0.05, ** *p* < 0.01, *** *p* < 0.001, **** *p* < 0.0001. 560 vs. 420, + *p* < 0.05, ++++ *p* < 0.0001. 2D vs. 3D, # *p* < 0.05, ## *p* < 0.01, #### *p* < 0.0001.

**Figure 4 ijms-21-06913-f004:**
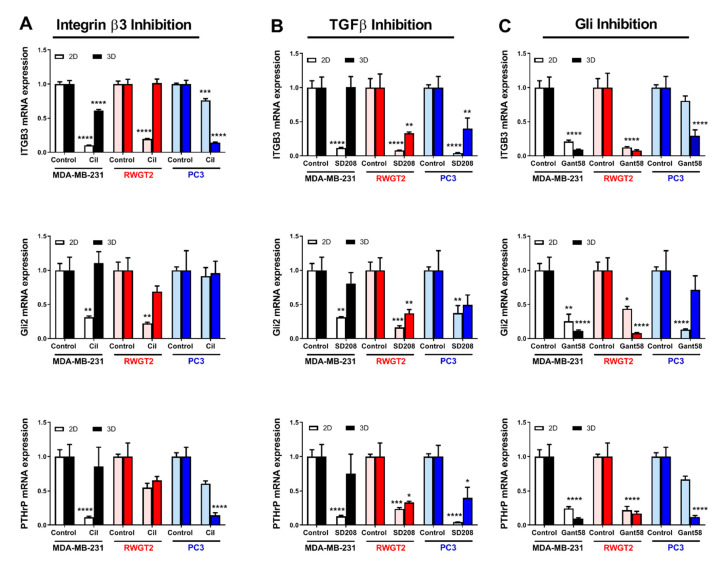
Effects of 3D culture on tumor cell drug response. Effects of 3D culture on inhibition of MDA-MB-231 (white—2D scaffolds, black—3D scaffolds), RWGT2 (pink—2D scaffolds, red—3D scaffolds), and PC3 (light blue—2D scaffolds, dark blue—3D scaffolds) tumor cells was tested for (**A**) Integrin β3 inhibition using the inhibitor Cilengitide (Cil), (**B**) TGF-β inhibition using an RI kinase inhibitor (SD208), and (**C**) Gli2 inhibition using the Gli antagonist (Gant58). Cells were grown on 2D rigid films (pale bars) or 420R 3D scaffolds (dark bars) and mRNA expression was analyzed by qRT-PCR after 48 h in culture. *ITGB3*, *Gli2*, and *PTHrP* mRNA expression decreased in response to SD208 and Cilengitide treatment in 2D films, but to a lesser extent in 3D scaffolds. Gli inhibition was effective in both 2D and 3D and all three tumor cell lines. Data presented as fold change over untreated control. Two-way ANOVA. 3D scaffolds/2D scaffolds treatment vs. control, * *p* < 0.05, ** *p* < 0.01, *** *p* < 0.001, **** *p* < 0.0001.

**Figure 5 ijms-21-06913-f005:**
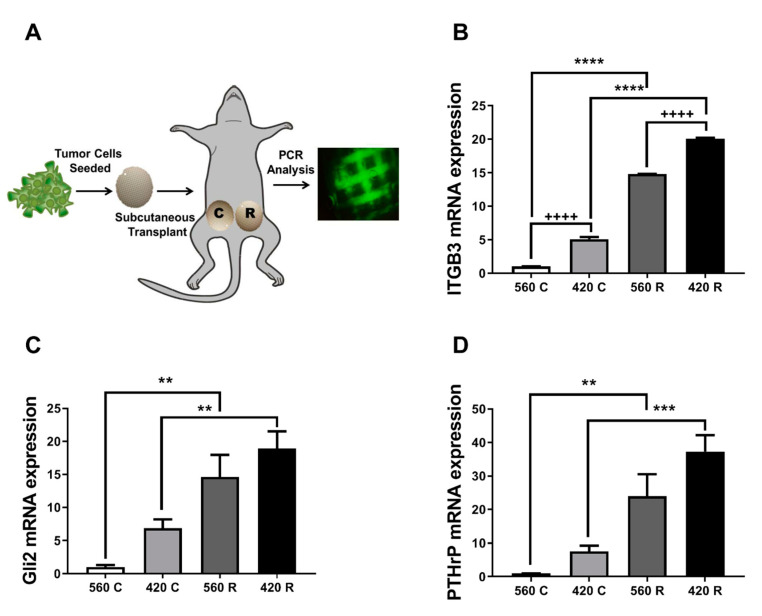
Effects of substrate modulus and pore size on bone metastatic gene expression in a xenograft model. (**A**) Schematic of animal model. Scaffolds were seeded with 1 × 10^6^ MDA-MB-231-GFP cells and implanted subcutaneously in the abdomen. After 3.5 weeks, scaffolds were harvested for qRT-PCR analysis to assess (**B**) *ITGB3*, (**C**) *Gli2*, and (**D**) *PTHrP* mRNA expression. mRNA expression was significantly upregulated on 420R scaffolds as compared to 560C 3D scaffolds. Data presented as fold change over 560C. Two-way ANOVA. Compliant vs. rigid, ** *p* < 0.01, *** *p* < 0.001, **** *p* < 0.0001. 560 vs. 420, ++++ *p* < 0.0001.

**Figure 6 ijms-21-06913-f006:**
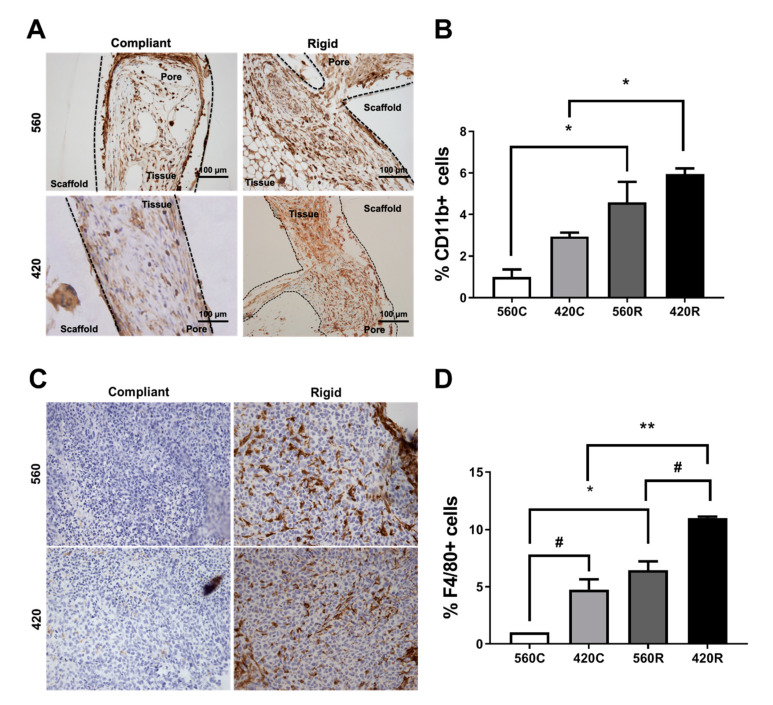
Effects of substrate modulus and pore size on immune infiltration in a xenograft model. Representative immunohistochemistry images show scaffolds with pores outlined by dashed lines. Tissue infiltrated inside pores is shown by Hematoxylin staining, and positive immunohistochemical staining is represented by dark brown staining. (**A**) Representative 20× images and (**B**) quantification of immunohistochemistry staining for CD11b in 3D scaffolds show increased CD11b+ cell populations in 560R (5-fold) and 420R (6-fold) scaffolds compared to compliant scaffolds. (**C**) Representative 20× images and (**D**) quantification of immunohistochemistry staining for F4/80 in 3D scaffolds reveal increased populations of F4/80+ cells in 560R (7-fold) and 420R (10-fold) as well as a noticeable increase in staining with decreasing pore size. All data was normalized to 560C. Compliant vs. rigid * *p* < 0.05, ** *p* < 0.01. 560 vs. 420, # *p* < 0.05.

**Figure 7 ijms-21-06913-f007:**
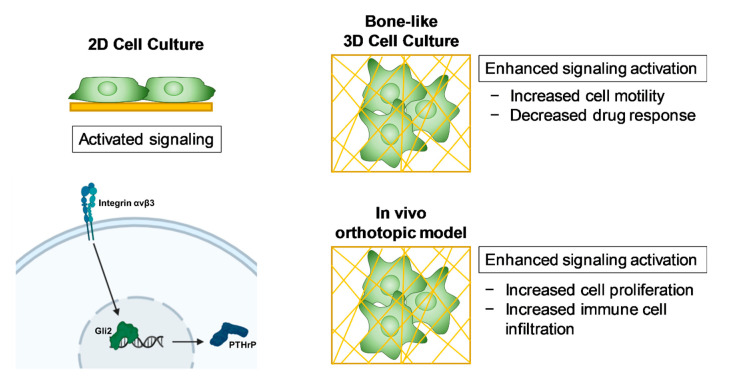
Schematic summary of the 3D bone-like culture system. Bone-like 3D culture of metastatic tumor cells increases signaling activation of genes associated with TIBD, increases cell motility, and influences immune cell infiltration in an in vivo orthotopic model.

**Table 1 ijms-21-06913-t001:** 3D scaffold characterization and in vitro study design. Material characterization and nomenclature of 2D films and 3D scaffolds.

Treatment Group	Label	Pore Size (μm)	Bulk Substrate Modulus *K*_s_ (MPa)
2D compliant film	2DC	N/A	5 ± 0.4
2D rigid film	2DR	N/A	266 ± 27
3D compliant scaffolds (small pores)	420C	423 ± 34	5 ± 0.4
3D compliant scaffolds (large pores)	560C	557 ± 44	5 ± 0.4
3D rigid scaffolds (small pores)	420R	423 ± 34	266 ± 27
3D rigid scaffolds (large pores)	560R	557 ± 44	266 ± 27

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
