# Peer review of "3D Bone Morphology Alters Gene Expression, Motility, and Drug Responses in Bone Metastatic Tumor Cells"

_ijms, 2020, doi:10.3390/ijms21186913_

Round 1

Reviewer 1 Report

This study addresses a really relevant aspect of bone metastasis: how the structure and mechanical forces of bone influences tumor cell colonization of bone microenvironment. Few studies are available on this topic, thus the presented data are original. The Authors propose an effective 3D bioprinted scaffold to study the interaction between cancer cell and bone. The methodology is correct and the results have been well described. I believe that this manuscript is suitable for publication.

Author Response

We thank reviewer 1 for their positive reviews.

Reviewer 2 Report

In this study Dadwal et al. present a 3 D model of bone to be utilized for the study of tumor- induced bone diseases (TIBD). It could be a good model system to study tumor- induced bone diseases.

I suggest the authors to revise this manuscript by addressing the following points:

  1. Cellular adhesion is an important step in the establishment of bone metastasis. Therefore, this scaffold needs to be measured for cellular adhesion by a quantitative assay.
  2. Since bone metastasis varies from being osteolytic to osteoblastic or more often mixed lesions, this scaffold could be tested for the difference in growth/viability of cancer cells that are known to form osteoblastic lesions vs lytic lesions.
  3. Since the bone microenvironment cells play an important role in bone metastasis, if would be pertinent to see whether the cancer cells growing on this scaffold start to expresses bone related markers, a phenomenon often referred to as osteomimicry.

Author Response

  1. Cellular adhesion is an important step in the establishment of bone metastasis. Therefore, this scaffold needs to be measured for cellular adhesion by a quantitative assay.

Performing standard adhesion test (like rolling assays, etc) are currently not possible on these 3D scaffolds, but we hope to develop more quantitative assays in the future. In the first version of the manuscript we included supplemental movies to demonstrate that the cells are attached, but motile, on the scaffolds. In other studies we have shown that that cells adhere to the fibronectin coating of the scaffold. As a surrogate to physical adhesion measurements we measured changes in gene expression of genes associated with tumor attachment and adhesion to the environment including FAK. These results have been added during this revision.

PG 3 LN 100: “Previous studies have shown expression of Collagen-1 and Fibronection in MSCs cultured on 3D scaffolds [28]. Here, we demonstrate changes in expression of proteins associated with tumor attachment and adhesion to the environment including Focal Adhesion Kinase (FAK) and Integrin Beta 3 (Ib3). FAK expression increases on MDA-MB-231 cells grown on rigid scaffolds compared to compliant scaffolds as shown by increased number of focal adhesions per cell (Supplemental Fig. S3).”

  1. Since bone metastasis varies from being osteolytic to osteoblastic or more often mixed lesions, this scaffold could be tested for the difference in growth/viability of cancer cells that are known to form osteoblastic lesions vs lytic lesions.

We agree that this could be a fantastic model for comparing osteolytic, osteoblastic, and mixed lesions. Especially, due to the limited number of models that can be used to study osteoblastic metastases this approach may allow for alternative approaches. However, since these tumors behave differently in vivo and in vitro, we focused on osteolytic tumors for these studies. Additional text has been added to the discussion to address this point. On Pg 11 starting at line 293 we added “Although these studies were performed with cells that primarily form osteolytic lesions, the 3D scaffolds described herein also have potential to be used to study tumors such as prostate cancer that form osteoblastic lesions”

  1. Since the bone microenvironment cells play an important role in bone metastasis, if would be pertinent to see whether the cancer cells growing on this scaffold start to express bone related markers, a phenomenon often referred to as osteomimicry.

We performed detailed analyses on many different genes associated with bone metastases. We agree that including this information is important for this manuscript. Thus, we have added a table in supplemental data (Table S.1) to show some these changes. In the text of the manuscript we added the following sentence on Pg 8 starting at line 212.

“We also determined via a Tumor Metastatic panel several genes associated with bone remodeling were highly expressed in the 3D rigid scaffolds when compared to the complaint scaffolds (Supplemental Table S.1).”